# Deep Learning and Structure-Based Virtual Screening for Drug Discovery against NEK7: A Novel Target for the Treatment of Cancer

**DOI:** 10.3390/molecules27134098

**Published:** 2022-06-25

**Authors:** Mubashir Aziz, Syeda Abida Ejaz, Seema Zargar, Naveed Akhtar, Abdullahi Tunde Aborode, Tanveer A. Wani, Gaber El-Saber Batiha, Farhan Siddique, Mohammed Alqarni, Ashraf Akintayo Akintola

**Affiliations:** 1Department of Pharmaceutical Chemistry, Faculty of Pharmacy, The Islamia University of Bahawalpur, Bahawalpur 63100, Pakistan; mubashirali035@gmail.com; 2Department of Biochemistry, College of Science, King Saud University, P.O. Box 22452, Riyadh 11451, Saudi Arabia; szargar@ksu.edu.pk; 3Department of Pharmaceutics, Faculty of Pharmacy, The Islamia University of Bahawalpur, Bahawalpur 63100, Pakistan; naveed.akhtar@iub.edu.pk; 4Department of Chemistry, Mississippi State University, Starkville, MS 39759, USA; abdullahiaborodet@gmail.com; 5Department of Pharmaceutical Chemistry, College of Pharmacy, King Saud University, P.O. Box 2457, Riyadh 11451, Saudi Arabia; 6Department of Pharmacology and Therapeutics, Faculty of Veterinary Medicine, Damanhour University, Damanhour 22511, AlBeheira, Egypt; gaberbatiha@gmail.com; 7Laboratory of Organic Electronics, Department of Science and Technology, Linköping University, SE-60174 Norrköping, Sweden; drfarhansiddique@gmail.com; 8Department of Pharmacy, Royal Institute of Medical Sciences (RIMS), Multan 60000, Pakistan; 9Department of Pharmaceutical Chemistry, College of Pharmacy, Taif University, P.O. Box 11099, Taif 21944, Saudi Arabia; m.alqarni@tu.edu.sa; 10Department of Biomedical Convergence Science and Technology, Kyungpook National University, Daegu 41566, Korea; ashraf.akintola@gmail.com

**Keywords:** NEK7, virtual screening, DFTs, deep learning, molecular dynamics, drug design, drug repurposing, structural-based, cancer

## Abstract

NIMA-related kinase7 (NEK7) plays a multifunctional role in cell division and NLRP3 inflammasone activation. A typical expression or any mutation in the genetic makeup of NEK7 leads to the development of cancer malignancies and fatal inflammatory disease, i.e., breast cancer, non-small cell lung cancer, gout, rheumatoid arthritis, and liver cirrhosis. Therefore, NEK7 is a promising target for drug development against various cancer malignancies. The combination of drug repurposing and structure-based virtual screening of large libraries of compounds has dramatically improved the development of anticancer drugs. The current study focused on the virtual screening of 1200 benzene sulphonamide derivatives retrieved from the PubChem database by selecting and docking validation of the crystal structure of NEK7 protein (PDB ID: 2WQN). The compounds library was subjected to virtual screening using Auto Dock Vina. The binding energies of screened compounds were compared to standard Dabrafenib. In particular, compound **762** exhibited excellent binding energy of −42.67 kJ/mol, better than Dabrafenib (−33.89 kJ/mol). Selected drug candidates showed a reactive profile that was comparable to standard Dabrafenib. To characterize the stability of protein–ligand complexes, molecular dynamic simulations were performed, providing insight into the molecular interactions. The NEK7–Dabrafenib complex showed stability throughout the simulated trajectory. In addition, binding affinities, pIC50, and ADMET profiles of drug candidates were predicted using deep learning models. Deep learning models predicted the binding affinity of compound **762** best among all derivatives, which supports the findings of virtual screening. These findings suggest that top hits can serve as potential inhibitors of NEK7. Moreover, it is recommended to explore the inhibitory potential of identified hits compounds through in-vitro and in-vivo approaches.

## 1. Introduction

Cancer is the most common cause of death, with a high mortality rate worldwide causing 10 million fatalities per year. Cancer is characterized by unregulated cell growth and rapid proliferation [1]. Uncontrolled cell proliferation, aggregation, and an aberrant cell cycle are hallmarks of human cancer. Typically, cell division is controlled by several regulatory factors, including protein kinases [2]. Among all known protein kinases, NIMA (never in mitosis, gene A) related kinase7 (NEK7) plays a multifunctional role [3], including centrosome duplication, intracellular protein transport, mitotic spindle assembly, DNA repair, and cytokinesis [4,5,6,7].

NEK7 is a highly conserved serine/threonine kinase consisting of approximately 302 amino acids [6]. NEK7 is structurally related to NEK6, which shares 85% amino acid sequence identity. However, NEK7 is involved in critical roles that NEK6 cannot take over. NEK7 is centrosome-localized and is known to be highly expressed in a variety of vital organs such as the heart, lung, fat, brain, liver, and spleen [8]. It enhances the centrosome duplication efficiency by promoting the pericentriolar material at the centrosome during the S and G1 phases [3].

In addition, NEK7 also encourages the proliferation of resting cells, which indicates its high-level involvement in various cancer types, including non-small lung cancer, breast cancer, NLRP3-related inflammatory disease, and gastric cancer progression [9]. NEK7 also has a promising role in growth and survival. NEK9 regulates the activation of NEK7 during mitosis, which promotes spindle assembly, centrosome separation, and mitotic division of the cell [7].

Besides promoting the proliferation of various resting cells, NEK7 is also involved in the progression and development of fatal inflammatory diseases, including Alzheimer’s disease, auto-immune disorders, inflammatory bowel disease, gout, and tumor formation [10]. Researchers have reported the involvement of NEK7 in the activation of NLRP3 inflammasome via ROS species formation, lysosomal destabilization, and potassium efflux. Stimulation of inflammatory mediators by NEK7 induces fibrosis and diabetic retinopathy and leads to hepatic carcinoma [10]. In brief, any mutation or atypical expression of NEK7 leads to the development of cellular oncogenesis and may provoke a fatal inflammatory response, causing tumorigenesis of multiple organs. These findings lend testimony to the involvement of NEK7 in the progression and development of numerous deadly diseases.

NEK7 is a promising target for multiple diseases, primarily cancer-related therapy research. NEK7 came into consideration two decades ago [2], but it has yet to be explored as a therapeutic target for preventing and treating NEK7-related diseases. A few medications have recently been developed to target the NEK7-mediated inflammasome pathway, but the mechanism and treatment outcomes are not specific and consistent [2].

Moreover, there is no FDA-approved medication that can selectively inhibit the expression of NEK7. Only Dabrafenib has shown activity against BRAF-mutant melanoma, which expresses more NEK9 [1]. These findings indicate that no published work has reported the selective and potent inhibitors of NEK7. As a result, the current study seeks more specific inhibitors that will provide a beneficial treatment option for NEK7-related cancer malignancies.

The current study focused on structure-based virtual screening (SBVS) of 1200 compounds library and drug repurposing of FDA-approved drug Dabrafenib. Dabrafenib demonstrated inhibitory potential against NEK9 with an IC50 value ranging from 1–9 nM [11,12]. Dabrafenib is comprised of benzene sulphonamide scaffolds. The basic sulphonamide group occurs in numerous biological active compounds [12], including anti-microbial [13], anti-tumor [14], anti-thyroid [15], antibiotics [16], and carbonic anhydrase inhibitors [17].

Clinically, sulphonamide-possessing drugs are used to treat lower urinary tract infections, whereas aromatic or hetero-aromatic sulphonamide derivatives possess a wide range of biological activities, including anti-tumor, anti-rheumatic, anti-microbial, and anti-inflammatory [18,19,20,21]. These findings have provided a strong rationale to retrieve structural analogues of Dabrafenib containing a basic sulphonamide nucleus.

The library of 1200 structural analogues of Dabrafenib was retrieved from the PubChem database and subjected to the in-silico drug discovery process. The discovery of a new anti-cancer agent is an extensive and laborious process. Thus, computer-aided drug design (CADD) [22] methods could serve as an alternative drug development strategy [23]. Among in-silico approaches, drug repurposing is an advanced tool for revisiting the activities of already approved drugs [24], which can save time and money [25].

The current study was focused to revisit the activity of Dabrafenib against NEK7 protein [26]. In addition, structure-based virtual screening (SBVS) [26] of 1200 structural analogues of Dabrafenib was carried out using molecular docking [27] and deep learning models [28]. SBVS is an advanced technology for the identification of potential hits with significant pharmacological properties against multiple molecular targets. Several robust docking programs are available for docking purposes in commercial and academic settings. In the present study, the Auto Dock Vina was used for virtual screening [29,30]. Moreover, density functional theory studies were conducted to explore the chemical reactivity profile of top-ranked analogues obtained through virtual screening. The structural geometry optimization and frequency calculations were performed. In addition, frontier molecular orbital (FMO) analysis and global reactivity descriptors were also determined. The efficacy of any drug is determined by its interaction with targeted biomolecules. Deep learning algorithms [31] were used for prediction of binding affinity and pIC_50_ values of top hits obtained via virtual screening. Predicted values of top hits were compared to in-vitro activity of Dabrafenib.

Furthermore, in-silico ADMET properties were also determined using a message-passing neural network (MPNN). The MPNN model is widely used for prediction of molecular properties such as blood brain barriers, human intestinal absorption, and solubility profiles [32]. The molecular docking approach only provides a static view of the molecular interactions of the complex. Still, to determine the stability of the protein–ligand complex, molecular dynamic simulations (MD simulations) have been performed to determine the stability, which provide significant insight into the molecular interactions of top-ranked complexes under accelerated conditions. Top hits obtained through structure based virtual screening a shown in Figure 1. All hits shared the same pharmacophore with standard Dabrafenib.

This is the first comprehensive computational study for the identification of selective inhibitors of the NEK7 protein. The current study has utilized the latest computational approaches, suggesting identified hits as a new strategy for treatment of NEK7-associated malignancies. Findings of the current study suggest the exploration of the inhibiting potential of these hits at the molecular level using in-vitro and in-vivo experimental techniques.

## 2. Experimental

### 2.1. Computational Studies

#### 2.1.1. Density Functional Theory Calculation

The ground state electronic energy is ascertained by electron density of the compound [33]. The electron density defines the number of electrons, nuclear charge and position of the nuclei in a compound [34]. Variation in electron density yields different ground state energy, and both of these properties are related by density functional theory methods [35]. DFT methods are based on suggestions that electron density can be accurately assumed by the set of specific orbitals using an exchange correlation such as B3LYP [36]. Based on their computational accuracy, DFT methods are a reliable and efficient approach for correct estimation of electronic properties of the compound [37]. The structural geometries of selected compounds were optimized through DFT studies. DFT calculations were executed on Guassian09 program [38] using B3LYPfunctional correlation and 6-31G* as a basis set [39]. It is a compelling theory to calculate the electronic structure of atoms and molecules. Gauss View 6 was used for visualization of output files [40]. In addition, DFT/B3LYP method was employed for generation of Frontier molecular orbitals (FMOs), electrostatic surface potential map and global and local reactivity of descriptors. After completion of calculations, the output log file was visualized in Gauss View 6 for determination of optimization energy, dipole moment, frequency and polarizability [41].

#### 2.1.2. Structure Based Virtual Screening (SBVS)

Drug candidates were retrieved from the PubChem database (https://pubchem.ncbi.nlm.nih.gov/) (accessed on 28 April 2022) to create the ligand library. There were 1200 structural analogues of Dabrafenib in the library. PyRx software was used to prepare the compounds library, which was converted to pdbqt format for virtual screening using Auto Dock Vina. The MMFF94 force field was used to minimize the energy of ligands. The crystallographic structure of the targeted protein was retrieved from Protein Data Bank (https://www.rcsb.org/) (accessed on 1 May 2022) (PDB ID: 2WQN). After that, MGL tools were used to prepare macromolecule, which included removing Het atoms and water molecules, and the addition of polar hydrogen. The protein was examined for any missing residues. Furthermore, Kollman’s charges were used to neutralize protein, and Gasteiger charges were calculated. Finally, for virtual screening of the compound library using Auto Dock Vina, a 1-angstrom grid box was built centered on the crystalline structure of protein at the point of co-crystal ligand (ADP) binding-site coordinates. The central xyz axis of the grid box was set to 80 × 80 × 80. Virtual screening was carried out after the targeted protein was prepared utilizing Auto Dock Vina’s script-based technique. The exhaustiveness was set to 5 and the number of nodes was set to 20. The virtual screening was repeated twice to ensure the accuracy of docking results. In addition, docking protocol was validated by re-docking the co-crystal ligand with targeted protein. A RMSD value of less than 2 angstrom indicates the reliability of the docking pose. After completion of virtual screening, the output findings of the virtual screening module were analyzed and docking scores of drug candidates were compared to standard Dabrafenib. Only four compounds were found to have higher docking scores than standard Dabrafenib. The top hits were subjected to further analysis using deep learning algorithms. Deep learning models were used to predict drug affinity and determine the stability of protein–ligand complexes.

#### 2.1.3. Molecular Dynamics Simulation

The molecular docking experiment provided an initial static protein–ligand complex for molecular dynamic studies. Desmond, a package from Schrödinger LLC [42], was used to run molecular dynamic simulation for 100 ns. Molecular docking studies provide insight into the binding state of ligand with protein. Docking produces the static orientation of a ligand molecule inside active pockets of targeted protein [43], and MD simulations measure the average displacement of atoms with respect to a reference. MD simulations provide information about the stability of the best complex [44,45].

Maestro or Protein Preparation Wizard were employed for processing of the protein–ligand complex. The system was prepared in the system builder tool of the Desmond package. The system was solvated by Monte-Carlo equilibration, TIP3P solvent model extended 10.0 angstrom in each direction. The counter NaCl ions at a concentration of 0.15 M were added to neutralize the system. The optimized potential for liquid simulation (OPLS 2005) [46] was used as a forcefield to generate parameter files [46]. The pressure control was conducted through the Martyna−Tuckerman−Klein chain coupling scheme with a coupling constant of 2 ps [47], whereas the Noose–Hoover chain coupling scheme was used for temperature control [48]. The energy minimization was performed for 20,000 steps in order to remove any intra-molecular steric clashes. Initially, the system was equilibrated (NVT ensemble) for 1 ns, and afterwards the NPT ensemble was performed for an additional 1 ns at 300 K temperature and 1 bar pressure. Finally, production run was performed for 100 ns under periodic boundaries conditions. The Particle Mesh Ewald (PME) method [49] was used to determine electrostatic interactions [50]. The Verlet/Leapfrog algorithm was used for numerical integration. A time step of 1 fs was used for minimization and a time step of 2 fs was used for molecular dynamic simulation [51]. Thermal MM-GBSA.py script [52,53] was used to calculate the ligand strain and ligand-binding free energy for docked conformations over a 100 ns period [54].

#### 2.1.4. Prediction of Binding Affinities, pIC_50_ and ADMET Properties Using Deep Learning Models

Dabrafenib, which has been approved by the FDA, has been found to be effective against BRAF-mutant melanoma with a high level of NEK9 protein expression. Dabrafenib’s inhibitory concentration was in the nanomolar range, 1–9 nM [1]. The drug’s effectiveness is largely determined by its binding affinity (IC_50_) and ADMET profile. Therefore, we have employed deep learning models to predict IC_50_, pIC_50_, and ADMET properties of top hits acquired through virtual screening in order to provide a direct comparison of binding affinities of top hits with standard Dabrafenib. Predicting the binding affinity and ADMET characteristics in silico, rather than using an experimental method, is a promising alternative. Deep learning (DP) models were used to predict drug target interactions (DTI) in the current work, which were formulated on encoder and decoder architectures. A DL model takes the SMILES string and amino acid sequence of the targeted protein as input and uses over 17 state-of-the-art DP learning techniques to predict drug efficacy indicators (Figure 2). The MPNN-CNN deep learning algorithms were used for affinity prediction in this work, while the MPNN model was used for ADMET predictions [32].

## 3. Results and Discussion

The 1200-compound library was retrieved from the PubChem database and subjected to SBVS and the FDA drug Dabrafenib. Dabrafenib was maintained as the standard drug to which docking scores of 1200 compounds were compared. It was observed that only four combinations have better docking scores and binding affinity than standard Dabrafenib. These four compounds were considered top hits and subjected to further analysis, including geometry optimization and FMO analysis via density functional theory studies. Moreover, IC50, pIC50, and ADMET properties of the top four compounds were also predicted using deep learning models.

### 3.1. Density Functional Theory Studies (DFTs)

Quickly calculating physicochemical properties of atoms, bonds and molecules is necessary to process thousands or millions of structures in data mining investigations. Calculations in quantum chemistry based on ab initio and density functional theory (DFT) yield increasingly reliable assessments of many characteristics [55]. The B3LYP hybrid functional is likely the most popular DFT functional, and its cost-effectiveness has been widely acknowledged. Nonetheless, DFT computations are still too computationally expensive to be conducted on single workstations or tiny clusters in less than a few hours [56].

#### 3.1.1. Optimized Geometries

In the present study, geometries of FDA-approved drug Dabrafenib and top hits were completely optimized using the DFT/B3LYP method and 6-31G* as a basis set. No negative frequencies were obtained after the geometry optimization, which demonstrates that current geometries are true local minima. Optimized structures of drug candidates are presented in Figure 3.

The compound **762** showed high value for polarizability and dipole moment, which indicates its high polarity and chemical reactivity. Optimization and polarizability values of top hits and Dabrafenib are given in Table 1.

#### 3.1.2. Frontier Molecular Orbital (FMOs)

The way a molecule interacts with other species is determined by its frontier molecular orbitals. The highest occupied molecular orbital, or HOMO, is the outermost orbital-bearing electrons, and it primarily works as an electron donor. The lowest unoccupied molecular orbital, or LUMO, is the innermost orbital with free electron acceptor sites. The ionization potential ought to be proportional to the HOMO energy, while the LUMO energy should be proportional to the electron affinity (Table 2).

The energy gap is the difference in energy between the HOMO and LUMO orbitals, and it is the most key variable in predicting the stability of a molecule. The HOMO–LUMO energy gap is used to evaluate the chemical reactivity and kinetic stability of the molecule. A soft molecule is a structure with a narrow gap that has a higher degree of polarization. As a measure of electron conductivity, the energy difference between HOMO and LUMO was recently employed to illustrate the bioactivity of intramolecular charge transfer (ICT). The stronger the chemical reactivity and the less stable the kinetics, the smaller the gap. The compound **762** has the narrowest energy gap at 0.127 eV among all the compounds. Compound **248** has a greater energy gap, measuring 0.152 eV. Thus, it shows compound **762** is chemically more reactive than all other compounds, which are comparatively the stable ones. In addition, the electron density of HOMO orbitals for Dabrafenib was localized over morpholine and piperdinyl rings, whereas electron density of LUMO is localized to the carbonitrile and benzocarbazole moiety of the drug. FMOs orbitals are shown in Figure 4.

#### 3.1.3. Global and Local Reactivity Descriptors

The HOMO and LUMO frontier orbitals are used to predict chemical reactivity. The HOMO orbital energy of a compound is significantly correlated with its vulnerability to electrophilic attack and ionization potential. A compound’s LUMO orbital energy is a reliable predictor of electron affinity and nucleophilic attack. The energy of LUMO is proportional to its electron affinity, indicating that it is susceptible to nucleophile attack. The frontier molecular orbital energies are also related to the hard and soft characteristics of a molecule. Hard nucleophiles have a low HOMO, whereas soft nucleophiles have a high HOMO. Similarly, hard electrophiles have a high LUMO energy, whereas soft electrophiles have a low LUMO energy. According to the frontier theory of electron reactivity, the chemical reaction occurs at the point where the HOMO and LUMO have the most overlap. All reactions require the HOMO density of the donor molecule, while all reactions require the LUMO density of the acceptor molecule. The frontier orbital densities of individual atoms can be used to quantify their reactivity inside a molecule. Chemical behavior is frequently predicted using electronegativity and hardness. Compound **248** presented greater energy gaps, indicating it to be the tougher among all compounds. Compound **208** demonstrated the greatest electrophilicity index value of 0.207 eV. This indicates that compound **208** is an excellent electrophile among all the other compounds. The HOMO–LUMO energy gap for Dabrafenib was found to be 0.159 eV. Dabrafenib showed a softness value of 6 (Table 3). The Koopman’s theorem was used to express ionization energy and electron affinity of drug candidates.
I = −EHOMO: A = −ELUMO

We evaluated the following parameters by using their respective formulas: Hardness: η = 1/2(ELUMO − EHOMO); Softness: S = 1/2η; Electronegativity: χ = −1/2(ELUMO + EHOMO); Chemical potential: µ = −χ; Electrophilicity index: ω = µ/2η.

### 3.2. Structure Based Virtual Screening and Predicted Binding Affinities

Initially, the Molecular docking methodology was validated by redocking a co-crystal ligand with targeted protein using the same coordinates. RMSD values of less than 2 angstrom were obtained, which demonstrate the successful validation of the docking protocol and can be used to describe ligand poses with specificity and accuracy. Afterward, virtual screening was conducted with 1200 ligands library and Dabrafenib. The coordinates of co-crystal ligand were used to dock the ligand library and Dabrafenib. Out of 1200 drug candidates, only four drug candidates showed excellent binding energies that were even better than Dabrafenib. The binding energies of top hits and Dabrafenib are tabulated in Table 4. In particular, compound **762** showed a maximum binding energy of −42.67 kJ/mol and exhibited strong binding affinity with NEK7 protein when subjected to the DL prediction model.

Among the four top hits, compound **208** exhibited promising hydrophobic and hydrophilic interactions. The amino acid residues involved in important molecular interactions were as follows: ASP115, ARG121, GLY117, ASP179, PHE168, ILE195, ALA114, ALA61, ILE40, and ASP118. It was observed that two hydrogen bonds were involved in stabilizing the protein–ligand complex. One hydrogen bond was observed with ASP118 with a bond length of 2.97 angstrom, while the second hydrogen bond was observed with GLY117 having a bond length of 3.14 angstrom. Important residues of the active site were engaged in hydrophobic interactions, including van der Waals interactions, pi-alkyl and alkyl–alkyl interactions. The docking score of compound **208** was found to be −33.47 kJ/mol. Similarly, compound **248** exhibited stronger molecular interactions with the following amino acid residues: ARG121, ASP118, ALA165, LYS63, ASN166, ASP179, PHE168, LEU111, VAL48, ALA116, ASP115, and GLY117. It was discovered that important amino acid residues of the NEK7 protein’s DLG/DFG motifs were involved in interactions. Furthermore, the amino acids LEU111 and LYS163 interacted via hydrophobic bonds. Two important hydrogen bonds were contributing toward the stability of conformations. One hydrogen bond engaged GLY117 residues with a bond length of 2.2 angstroms. Another hydrogen bond was engaging ASN166 amino acid with a bond length of 3.34 angstroms. Among hydrophobic interactions, van der Waals interactions played a pivotal role in stabilizing the complex. The docking score of the compound **248** was −35.56 kJ/mol. The putative 2D and 3D binding modes of compounds **208** and **248** are shown in Figure 5.

Another important top hit was compound **255**, which exhibited potent molecular interactions with amino acid residues of the active site. It was the second-best drug candidate obtained via virtual screening. Amino acid residues involved in bonding and nonbonding interactions were as follows: PHE45, SER46, LYS63, ALA114, ASP115, GLU112, PHE168, ASP179, VAL48, GLY43, and GLN44. It was observed that two important hydrogen bonds with short bond lengths were contributing toward stabilizing the complex. One hydrogen bond occurs between the electronegative oxygen atom of the compound **255** and the SER46 residue of the targeted protein. Moreover, the second hydrogen bond was engaged in GLY117 with a bond length of 3.16 angstroms. As shown in Figure 5, amino acid residues from the active site were involved in hydrophobic interactions with compound **255**.

Now referring to the top hit obtained through SBVS, namely compound **762**, It has shown excellent docking scores and demonstrated significant binding affinity obtained through deep learning models. It was observed that compound **762** was engaged in three hydrogen bonds of moderate-to-strong strength. One hydrogen bond occurred between the pentazole ring of the compound **762** and the electronegative oxygen atom of TYR201. The bond length of interaction was 3.08 angstroms. Similarly, the second hydrogen bond engaged SER234 residues with a surprisingly smaller bond length of 2.92 angstroms. These interactions lend enough testimony to stronger molecular interactions and more stabilized protein–ligand complexes. Furthermore, the third and last hydrogen bond occurred between TYR237 and compound **762** with a bond length of 3.02 angstroms. All three amino acid residues involved in hydrogen bonding belong to the activation loop of the NEK7 protein. The remaining active site residues, ILE123, GLU228, PHE236, MET203, PRO200, LEU246 and LEU232, engaged in hydrophobic interactions with compound **762**. The docking score and binding affinity (IC_50_) were predicted to be best among all top hits, i.e., −42.67 kJ/mol and 61.74 nM, respectively. Compound **762** could be a promising drug candidate for the treatment of NEK7-associated malignancies. The binding interactions of compounds **255** and **762** are shown in Figure 6.

The bonding and non-bonding interactions of standard Dabrafenib was involving important amino acid residues of NEK7 activation loop. ARG50, LYS38, ALA165, ILE40, GLY117, ASP115, PHE168, LEU111, LEU112, ALA114, LEU113, ALA161, ASP179, and ILE95 were the amino acid residues implicated in molecular interactions with Dabrafenib. Dabrafenib exhibited significant molecular interactions, which contributed towards complex binding affinity. The strong interactions were observed with targeted protein and sulphonamide rings. The sulphonamide ring was implicated in several important stabilizing contacts, including conventional hydrogen bonding with ASP115 of the activation loop, Pi-cation interaction with ARG50, and interactions with ILE40 and ASP115 by two fluorine atoms connected to the ring. PHE168 formed pi-cation and pi-pi T-shaped contacts with the butylthiazole ring, whereas the pyrimidine ring produced conventional hydrogen bonds with GLU112 and ASP179, a carbon–hydrogen connection with ALA114, and a pi-alkyl interaction with ALA161. Due to important chemical interactions, Dabrafenib has a good binding energy of −33.89 kJ/mol. van der Walls interactions are essential hydrophobic interactions that have been observed with the amino acids LYS38, ALA165, GLY117, LEU111, LEU113, and ILE95. Figure 7 depicts the probable binding mode of Dabrafenib with NEK7.

### 3.3. Electrostatic Surface Potential Map

Investigating the electrostatic surface potential (ESP) map is a key activity in drug design as it determines the chemical reactivity of the compound and its ability to produce important molecular interactions. It is an effective way to visualize the molecular reactivity and evaluate the nature of ligand-binding with a targeted protein. The ESP map is depicted by different colored regions depending upon the electronegativity of the compound. The highly electronegative part is represented by the color red, whereas the electropositive part is represented by the color blue. The QM calculations were performed using DFTs at the B3LYP/6-31G* level of theory. Figure 8 depicts the ESP potential and the nature of ligand-binding with the targeted protein. In this study, the contribution of the electronegative oxygen atom in all interactions is indicated by the color red, whereas the contribution of the nitrogen atom is provided in the color blue. Considering the electrostatic surface potential map, the contribution of oxygen atoms toward interaction potential is higher than that of nitrogen atoms. It was observed that in the case of compound 208, the electronegative oxygen atom was acting as a hydrogen acceptor and was producing strong hydrogen bonding with GLY117. Similarly, in compound **248**, **255** and **762**, electronegative oxygen atoms were involved in stronger intermolecular interactions. In contrast, nitrogen atom was involved in hydrogen bonding by donating the hydrogen bond for example, in case of compound **208**, nitrogen was donating hydrogen bond to ASP118 residue. In addition, the docked conformation of ligands on the protein surface is also represented by different colored regions (Figure 8). The red surface indicates the hydrogen bond acceptor region, while the blue surface indicates the hydrogen bond donor locations. Whereas the grey color areas indicate the hydrophobic interactions, including van der Waals interactions. The red-colored surface area of protein is buried by nitrogen atoms as they act as proton donors, whereas the blue-colored protein surface is buried by electronegative atoms such as oxygen, fluorine and chlorine, which acts as a hydrogen bond acceptor. It can be observed that the grey surface area of protein is mostly involved in hydrophobic interactions, and these regions are buried by alkyl, phenyl rings and other hydrophobic groups present in all compounds.

### 3.4. Buried Surface Area (BSA)

Molecular interactions are the critical factors in determining the stability of protein–ligand complexes. Molecular interactions existing between protein–ligand complexes can be modelled by taking into account the physicochemical properties and complementarity of the shape of the binding interface. A useful method for determining the complementarity of the shape and extent of molecular interactions is the estimation of the buried surface area (BSA) of a protein–ligand complex. In the current study, the BSA of best complexes was calculated using a new Shrake–Ruply algorithm-based tool (*dr_sasa*) [57] used for calculating the solvent accessible surface area (SASA), buried surface area (BSA), and contact surface area (CSA). All four top compounds (**208**, **248**, **255**, and **762**) were subjected to the calculation of BSA. It was observed that the targeted NEK7 protein was buried up to 80% and 70% by compounds **208** and **248,** respectively. In particular, amino acid residues ILE40 and PHE168 were strongly buried by compound **208** (49 Å^2^). Compound **248**, on the other hand, was strongly engaging the ARG121 and PHE168 with BSA of 39 Å^2^ and 41.3 Å^2^ respectively. The detailed buried surface area of both compounds is shown in Figure 9.

In terms of compounds **255** and **762**, it was observed that both compounds significantly engaged the amino acid residues of the target protein. Compound **255**, in particular, was burying the surface area of the NEK7 protein by up to 360 Å^2^. The BSA of compound **255** with VAL48, LYS63, ALA114, and PHE168 was 168, 212, 187, and 351 Å^2^, respectively, which was the best among all top hits. These values demonstrate the strong nature of molecular interactions existing between the target protein and compound **255**. In the case of compound **762**, important amino acid residues were buried by compound **762**. In particular, TYR201, TYR237, and MET241 were significantly buried by compound **762** with BSA of 64, 72, and 25.6 Å^2^. Moreover, it was worth noticing that the major contributing atoms were oxygen, nitrogen, fluorine, sulphur, and chlorine, which were involved in increasing the contact surface area of compounds with a targeted protein. The BSA of compounds **255** and **762** is shown in Figure 10.

### 3.5. Molecular Dynamic Simulation

The molecular docking technique is comparatively rapid and imprecise. The docking deficiencies and flexibility of protein may interfere with protein–ligand complex. However, molecular dynamic simulations are computationally expensive and time-consuming but provide reliable and accurate illustration of protein displacement. Considering these facts, molecular dynamic simulations were undertaken using Desmond software package [58,59]. The root-mean-square deviation (RMSD) patterns provide significant insight into average change in displacement of atoms with respect to a frame. The RMSD trajectory provides information about the structural configuration of protein. It is computed for each frame of the trajectory. In order to gain insight into the structure of a protein, it is important to monitor the protein’s RMSD. Plotting the RMSD of the ligand is possible once the protein–ligand complex is aligned on a reference protein backbone and the RMSD of the ligand heavy atoms is measured. It is likely that the ligand has diffused from its initial binding site if measured values exceed the protein’s RMSD by a substantial margin. Molecular dynamic trajectory analysis is also used to determine the root-mean-square fluctuation of the targeted protein.

#### 3.5.1. RMSD Analysis of Protein and Protein–Ligand Complexes

The RMSD patterns for C-alpha atoms of NEK7 protein were estimated in order to determine the effect of the bounded drug on the conformational stability of NEK7 protein. Figure 11 is displaying the progression of RMSD values for the C-alpha atoms of NEK7 as a function of time. The 2wqn–Dabrafenib complex reaches the equilibrium after around 5 nanoseconds of simulation, and although side chain residues displayed fluctuations, they remained in the permissible range of 1–4 angstroms, which can be considered insignificant [60]. The NEK7–Dabrafenib complex showed slight fluctuations after 50 ns, which again became stable after 60 ns of simulation and remained equilibrated throughout the simulated trajectory. RMSD fluctuation was observed from 70 to 90 ns, which is due to the decrease in the number of contacts during this time, but after 90 ns, the number of contacts with amino acid residues increased and RMSD pattern became stable. It demonstrate the existence of stable molecular interactions. After being equilibrated, NEK7 RMSD values fluctuated within 2 angstrom. After 80 ns, protein RMSD showed slight fluctuation up to 2.5 angstrom and dropped again after 95. The average RMSD value for the protein–ligand complex and NEK7 protein is tabulated in Table 5.

These findings suggest that the ligands stayed firmly bound to the receptor throughout the simulation period. Moreover, small RMSD patterns indicate the fewer structural re-arrangements and lesser conformational changes within binding site residues [61].

It is beneficial to identify local differences in the protein chain by using the root-mean-square fluctuation (RMSF). Figure 12 peaks on the RMSF graph represent the regions of the protein that change the most during the simulation. The average RMSF for the NEK7 backbone was 0.87 angstrom, indicating the fewer structural rearrangements (Table 5). The N- and C-terminal ends of proteins are more likely to undergo alteration than any other portion of the protein. In the range of amino acids from 180 to 220, the RMSF value fluctuated, as can be seen in the RMSF graph. These residues are found in the C-terminal lobe. The protein’s structure, such as its alpha and beta helices and strands, tends to be stiffer and less variable than its unstructured component. According to MD trajectories, the residues with the highest peaks are found in loop areas or the N- and C-terminal regions. Binding site residues with low RMSF values imply a stable ligand–protein interaction.

The contact profiles of NEK7–Dabrafenib were computed from simulated trajectories, as shown in Figure 13. FDA-approved drug Dabrafenib interacted with ILE40, LYS163 and ARG121 through Water Bridge and hydrogen bonding. The amino acid residues, ILE40, LYS163, and ARG42, were involved in H-bonding. During MD simulations, 12 hydrogen bonds were found to be dominant with significant occupancy. Details of hydrogen bonding is given in Table 6.

The hydrogen bond with ALA114 existed for more than 25% of simulation time. Hydrophobic interactions existed between VAL48, LEU113, VAL48, and PHE168. These molecular interactions contributed towards stabilizing the protein–ligand complex.

#### 3.5.2. Radius of Gyration (Rg) and Solvent-Accessible Surface Area of Protein (SASA)

Radius of gyration (Rg) is measure of protein compactness, stability, integrity and foldness of protein backbone. The Rg trajectory for NEK7 is depicted in Figure 14. Trajectory analysis for the radius of gyration revealed that protein retained compactness throughout the simulated trajectory, and only slight fluctuations were observed around 30 ns, which stabilized after a short period of time.

Solvent-accessible surface area (SASA) is the area of protein that is accessible by the solvent. The higher the value for SASA, the lower the stability of the protein. In the current study, residue wise SASA was calculated and ranged between 180 to 350 Å^2^, which is quite acceptable. The average SASA value was computed to be 282.72 Å^2^ (Table 5). The residue wise SASA of targeted protein is shown in Figure 15.

#### 3.5.3. Principle Component Analysis (PCA)

It is an essential multivariate statistical technique used to describe the protein dynamics in a spatial scales. It is a linear relationship that extracts essential features of protein using covariance and/or correlation matrices. These matrices are derived from the atomic coordinate that represents the accessible degree of freedom (DOF) of the protein in a simulated trajectory. In the current study, Pearson’s cross-correlation matrix was employed as it can normalize the large protein variables and prevent high atomic variations that can skew the results. In addition, eigenvectors with a specific variance value also play an important role in characterizing the motion of protein in spatial scales. In the current study, essential dynamics of protein were calculated by applying PCA analysis to the protein trajectory. It was observed that different variables were forming tight clusters with narrow angles, which indicates that they were correlated with vectors (PCs) [62]. PCs are the vectors that are used to describe protein motion with respect to variables. Two PCs are used in the current study to characterize the protein motion. In Figure 16, it can be observed that PC1 and PC2 are clearly indicating the behavior of various variables. Distribution on the scatter plot indicates the protein components are tightly clustered with small angles.

Correlation matrices are also the correlation coefficients between variables and PCs. In Pearson’s cross-correlation, the percent of variance in a protein variable is explained by PCs. Figure 17 is depicting the Pearson correlation graph for NEK7 variables.

#### 3.5.4. MM-GBSA Energy Calculations

Molecular docking is a robust technique for determining the binding orientation of a protein–ligand complex. However, it is still lacking in its ability to correctly identify the binding affinities of docked ligands. In order to determine correct binding energies of docked conformations, MM-GBSA energy calculations were performed, which are an efficient and reliable method for the determination of binding free energies. The MM-GBSA method provides free energy calculations by taking into account all hydrophobic, hydrophilic and electrostatic interactions [63]. After energy calculations, values obtained were more negative and showed stronger binding affinities, as compared to the docking scores obtained from molecular docking. The following equation was used to calculate binding free energy [64];
ΔG_bind_ = ΔE _mm_ + ΔG _sol_ + ΔG _SA_

The MM-GBSA energies for the protein–ligand complex was determined through the Thermal_mmgbsa script of Schrodinger. MM-GBSA energies are tabulated in Table 7.

#### 3.5.5. MM-PBSA Energy Calculations

In MMPBSA energy analysis, the free binding energies of protein, ligand and protein–ligand complex are estimated by following equation;
G = E_bnd_ + E_el_ + E_vdW_ + G_pol_ +G_np_ − TS
where, E_bnd I_ refers to bond energy, E_el_ refers to electrostatic energy and E_vdW_ represents van der Waals interactions. In the current study, Poisson–Boltzmann calculations were performed using the internal PBSA solver in mmpbsa_py_energy. All units are represented in kcal/mol. MM-PBSA energy analysis is given in Table 8.

### 3.6. ADMET Profile

In-silico ADMET properties of top-ranked hits were determined by deep learning models; more than 17 models were employed at the backend, which provided predictions on the ADME profile of each hit. It is an important part in drug development that can identify the desired pharmacological properties of compounds. In the current study, the message passing neural network (MPNN) is employed for the determination of ADMET properties. It was observed that compound **762** showed the lowest clinical toxicity value of 0.28%. The ADMET profile of top hits is tabulated in Table 9.

## 4. Conclusions

In the current study, structure-based virtual screening of a 1200-compound library and Dabrafenib was carried out using Auto Dock Vina. These compounds are in the early stages of drug development, and the in-silico approach used in this study was contributing toward investigating the inhibiting potential of these compounds through molecular docking, DFTs, and MD simulation, as well as determining the drug-like properties of these compounds through deep learning models. The FDA-approved drug, Dabrafenib, was considered as a standard drug to which in-silico findings could be compared. SBVS findings discovered four important hits having better binding energies as compared to standard Dabrafenib. In addition, the chemical reactivity profiles of top hits were determined via DFT studies. Findings from DFT studies revealed the reactive nature of the compounds. Moreover, the current study has utilized deep learning models for prediction of binding affinity, pIC_50_, and ADMET properties. It was observed that compound **762** showed good binding affinity and demonstrated a promising ADMET profile. Moreover, molecular dynamics simulations were performed to determine the stability of the protein–ligand complex under accelerated conditions. It was observed that the ligand remained significantly attached to the protein-activation loop, suggesting potential inhibiting activity of the compound. In short, the findings of the current study identify top hits that could prove an effective treatment strategy for NEK7-associated cancer malignancies. These findings will assist researchers to develop newer leads without consuming much time and money. Further experimental studies are also recommended for future prospects.

## Figures and Tables

**Figure 1 molecules-27-04098-f001:**
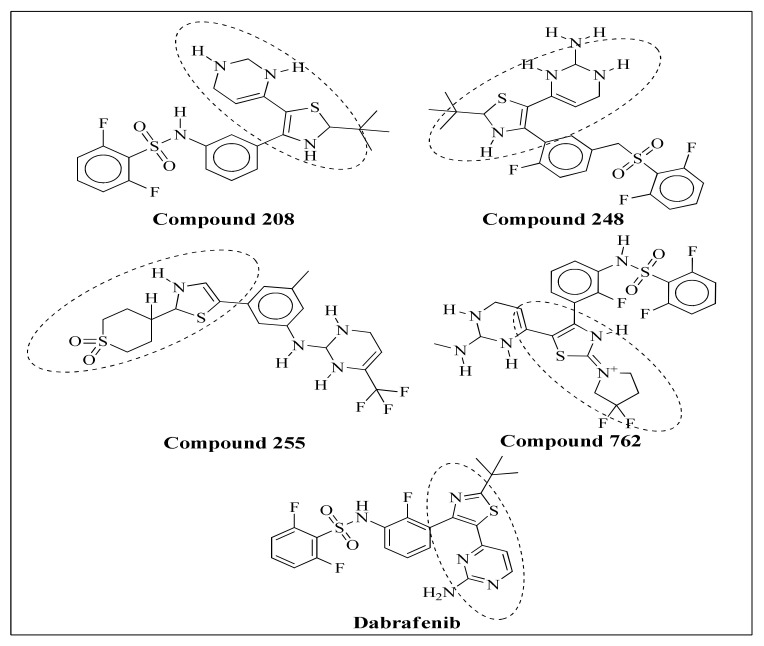
Top Hits obtained through SBVS. All hits were sharing same basic Pharmacophore with standard Dabrafenib.

**Figure 2 molecules-27-04098-f002:**
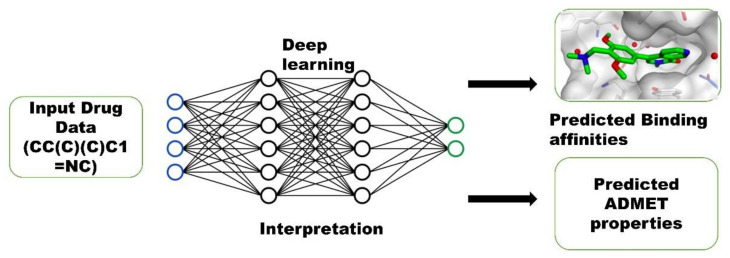
Implementation of Deep learning Model.

**Figure 3 molecules-27-04098-f003:**
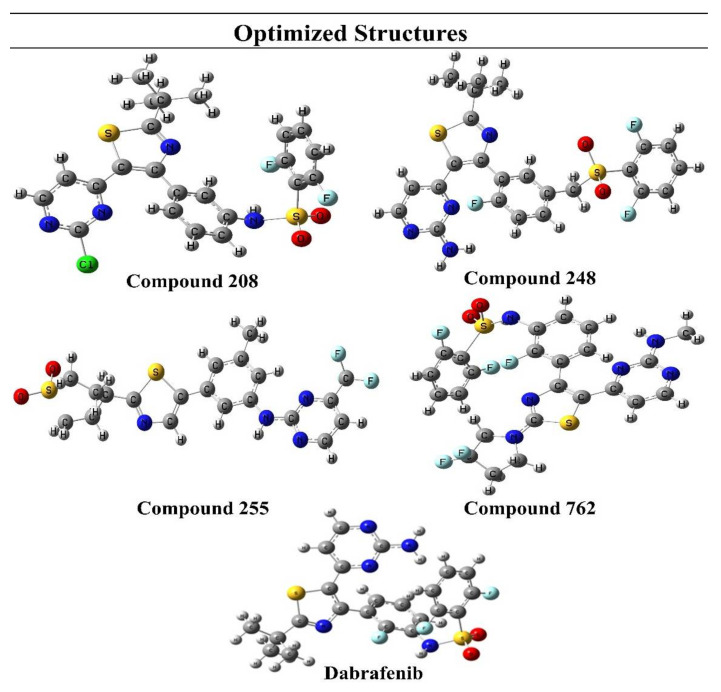
Optimized structures of selected compounds.

**Figure 4 molecules-27-04098-f004:**
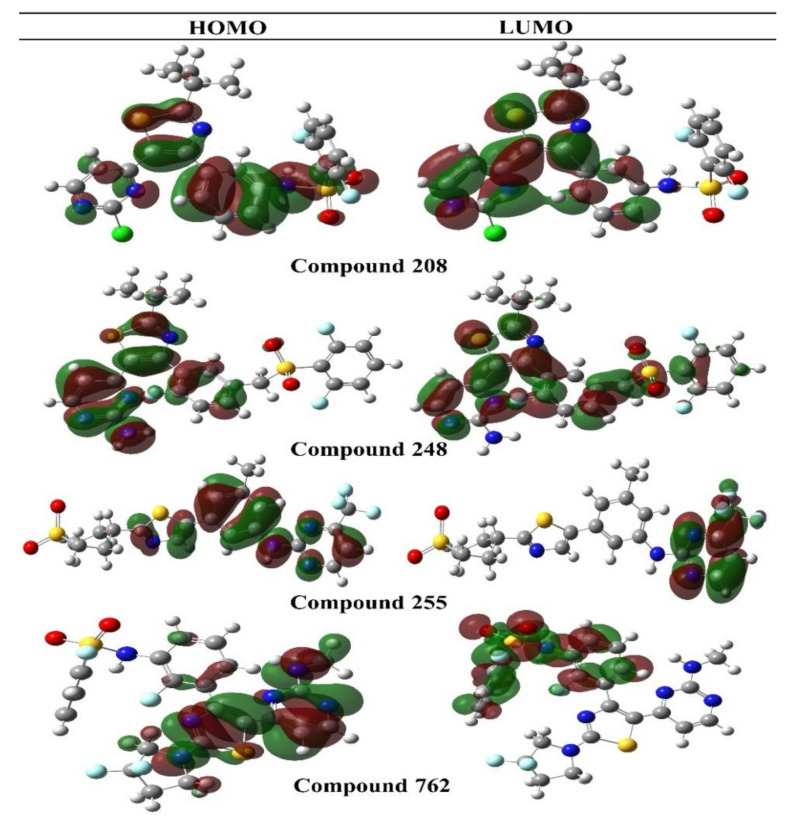
HOMO–LUMO structures of the selected compounds.

**Figure 5 molecules-27-04098-f005:**
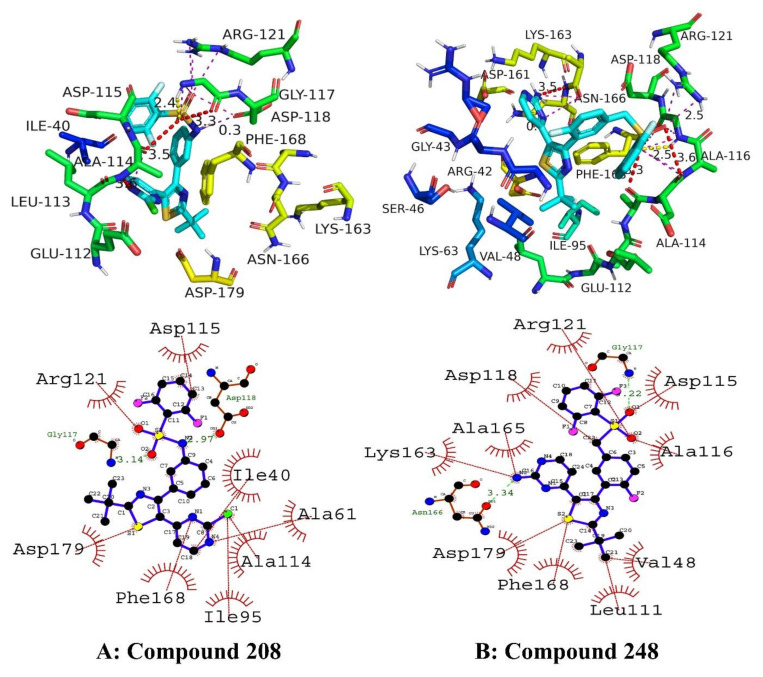
The putative 2D and 3D binding mode of compound **208** (**A**) and **248** (**B**). Green dashes are indicating hydrogen bonding whereas red dashes are indicating hydrophobic interactions.

**Figure 6 molecules-27-04098-f006:**
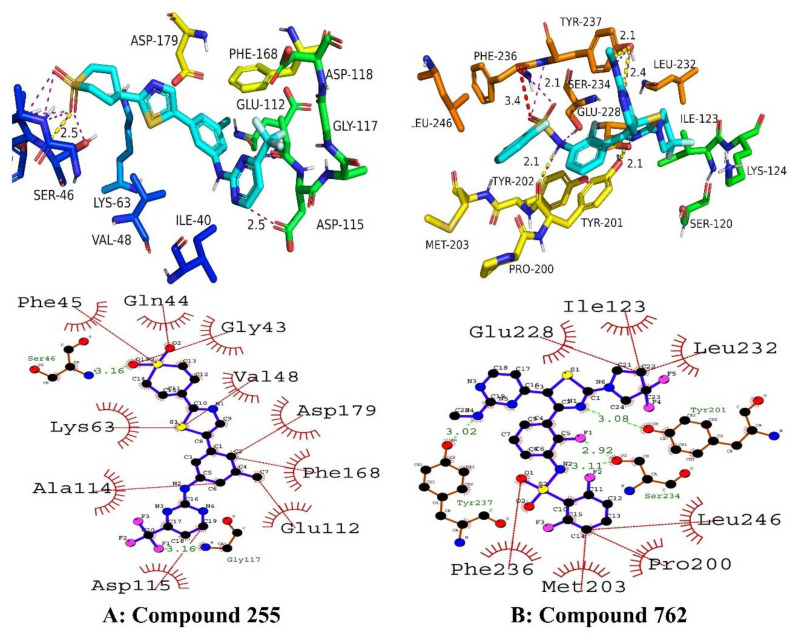
The putative 2D and 3D binding mode of compound **255** (**A**) and **762** (**B**). Green dashes are indicating hydrogen bonding whereas red dashes are indicating hydrophobic interactions.

**Figure 7 molecules-27-04098-f007:**
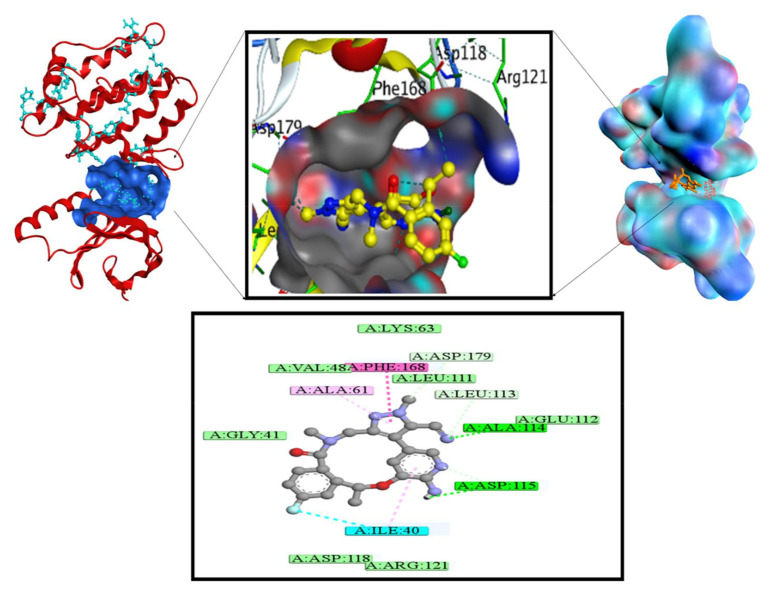
2D and 3D interactions of NEK7–Dabrafenib complex.

**Figure 8 molecules-27-04098-f008:**
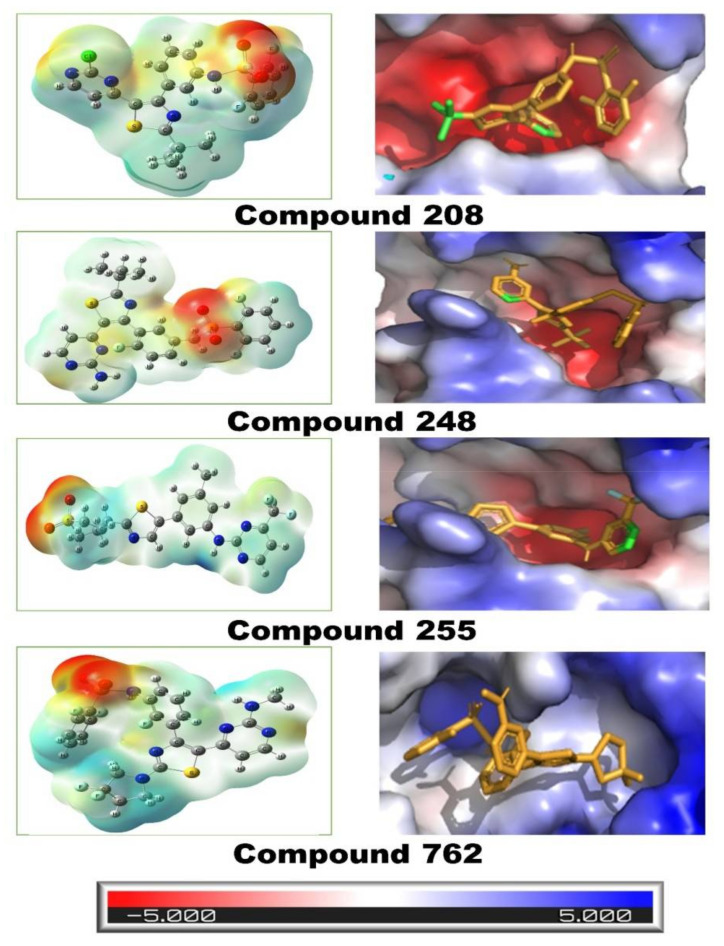
Electrostatic surface potential map of all ligand complexes.

**Figure 9 molecules-27-04098-f009:**
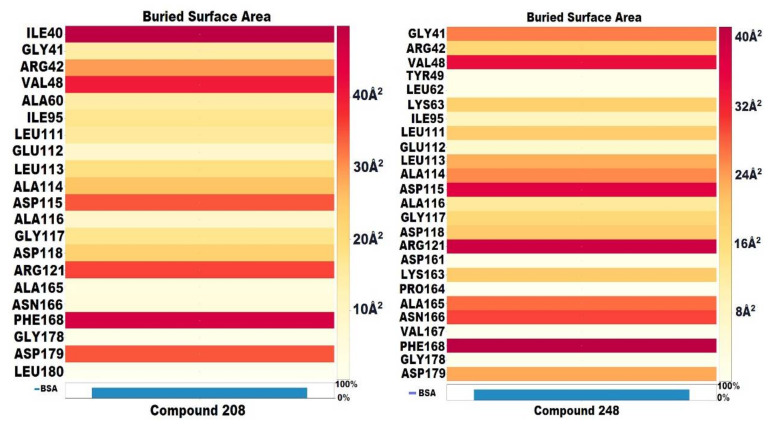
Buried surface area (BSA) of compound **208** and **248**.

**Figure 10 molecules-27-04098-f010:**
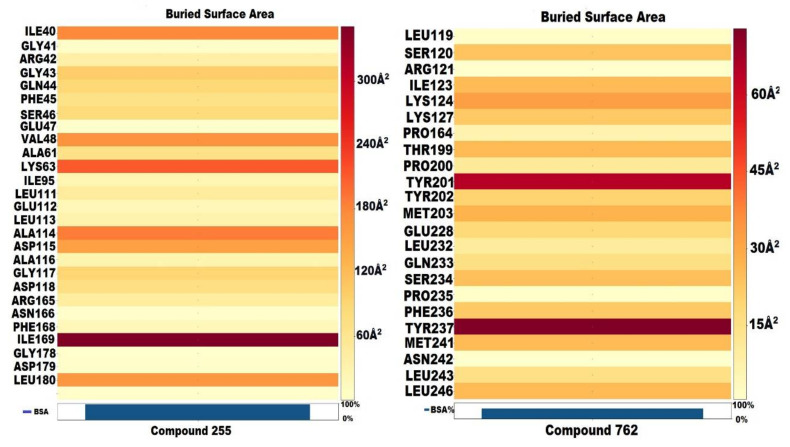
Buried surface area (BSA) of compound **255** and **762**.

**Figure 11 molecules-27-04098-f011:**
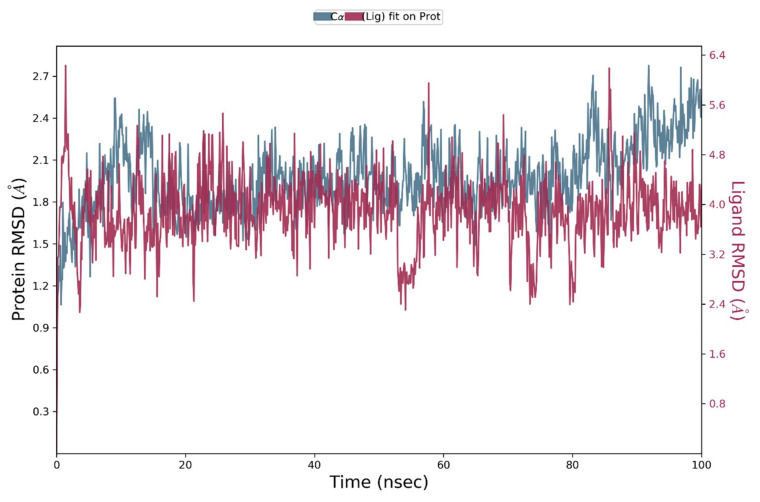
Residue wise root-mean-square deviation (RMSD) of the C-alpha atoms of NEK7 (2wqn) and Dabrafenib Complex.

**Figure 12 molecules-27-04098-f012:**
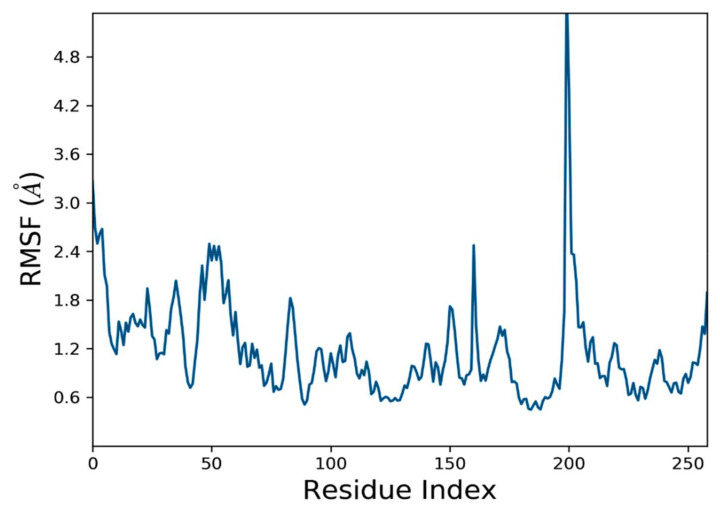
Root-mean-square fluctuations (RMSF) of the C-alpha atoms of NEK7 (2wqn).

**Figure 13 molecules-27-04098-f013:**
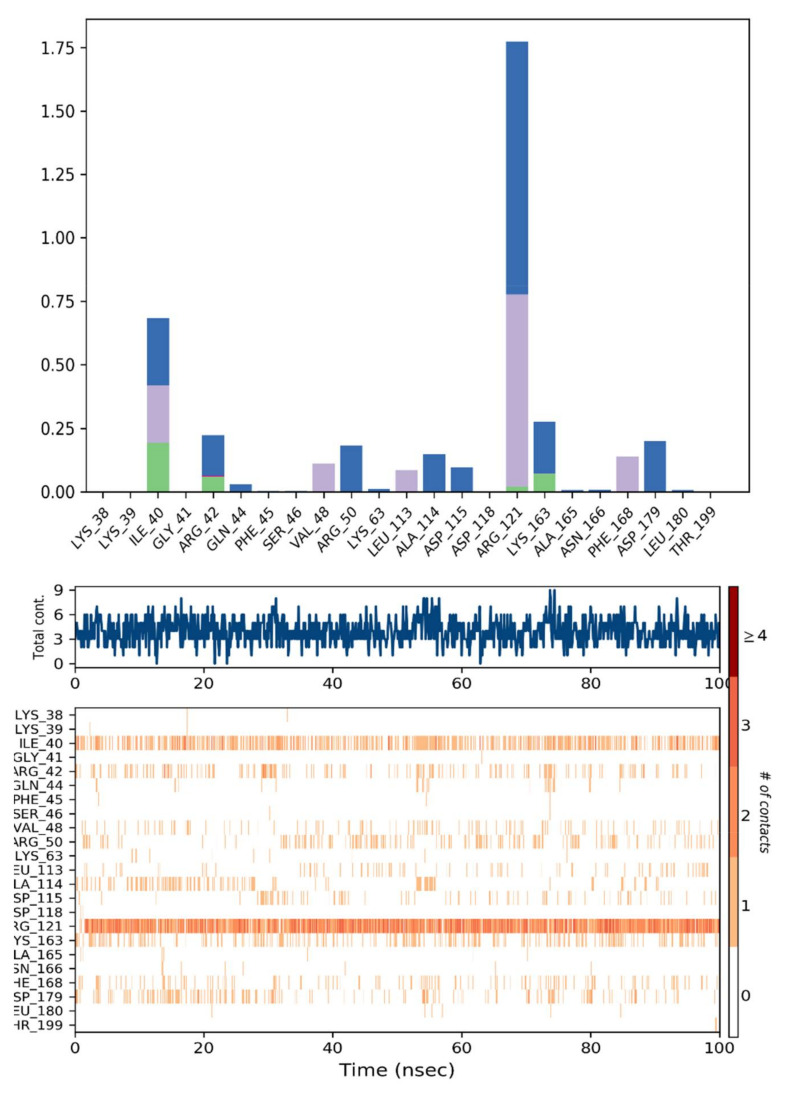
NEK7-Dabrafenib Contact histogram.

**Figure 14 molecules-27-04098-f014:**
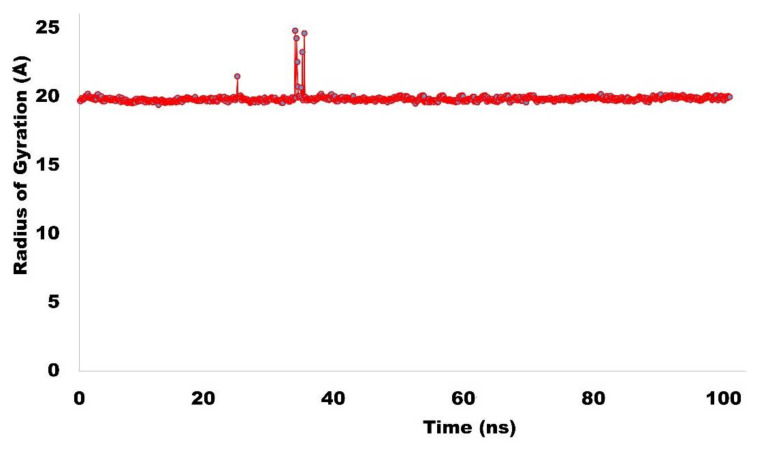
Radius of gyration (NEK7).

**Figure 15 molecules-27-04098-f015:**
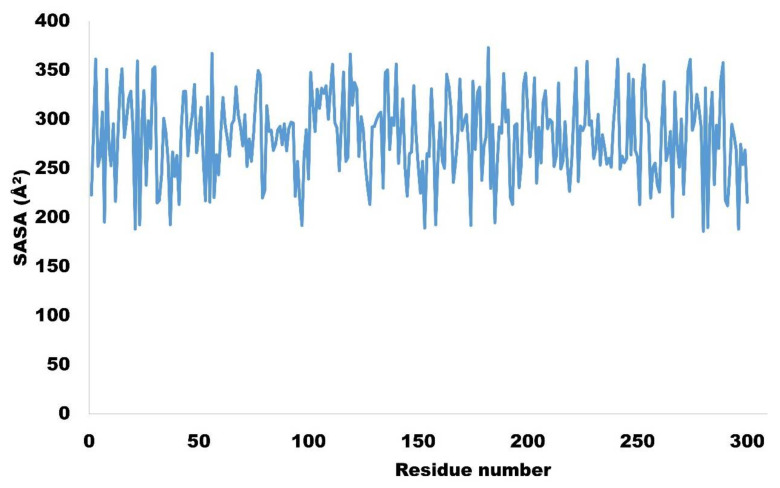
Residue wise solvent accessible surface area (SASA) for NEK7.

**Figure 16 molecules-27-04098-f016:**
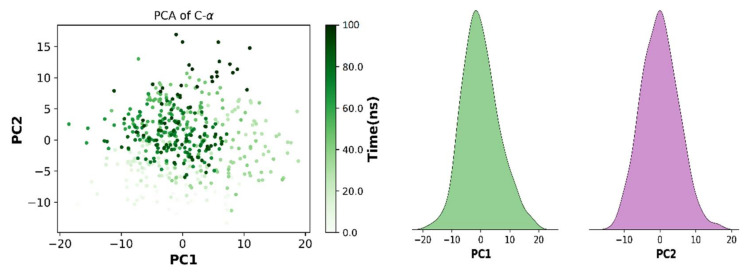
The correlation between protein variables and two top PCs.

**Figure 17 molecules-27-04098-f017:**
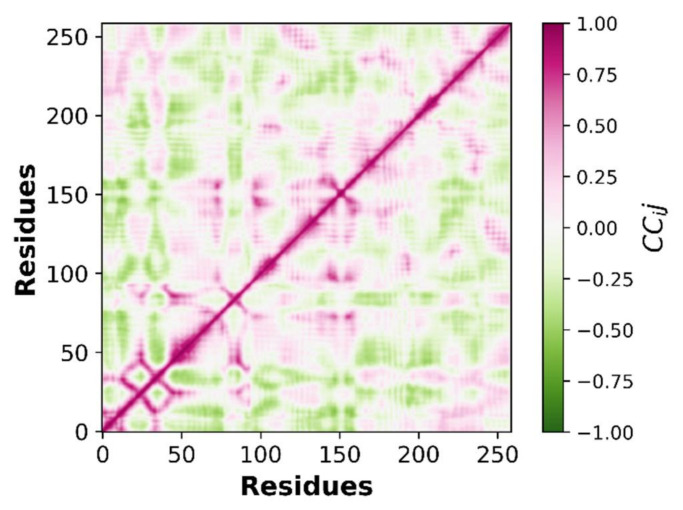
Pearson correlation graph for NEK7 variables.

**Table 1 molecules-27-04098-t001:** Energetic parameters of top hits and standard Dabrafenib.

Compound	Optimization Energy (Hartree)	Polarizability (α) (a.u.)	Dipole Moment (Debye)
Compound **208**	−2712.903	340.588	13.330
Compound **248**	−2391.895	345.671	7.283
Compound **255**	−2238.320	304.820	8.827
Compound **762**	−2699.752	360.213	10.042
Dabrafenib	−2407.203	319.254	6.682

**Table 2 molecules-27-04098-t002:** ΔE_gap_ of HOMO/LUMO orbitals of selected compounds.

Compound	E_HOMO_(eV)	E_LUMO_(eV)	∆E_gap_(eV)	Potential Ionization I (eV)	Affinity A (eV)
Compound **208**	−0.234	−0.099	0.135	0.234	0.099
Compound **248**	−0.222	−0.070	0.152	0.222	0.070
Compound **255**	−0.228	−0.083	0.145	0.228	0.083
Compound **762**	−0.216	−0.089	0.127	0.216	0.089
Dabrafenib	−0.233	−0.074	0.159	0.233	0.074

**Table 3 molecules-27-04098-t003:** Global reactivity descriptors.

**Compound**	**Hardness** **(η)**	**Softness (S)**	**Electronegativity (X)**	**Chemical Potential (μ)**	**Electrophilicity Index (ω)**
Compound **208**	0.067	7.433	0.167	−0.167	0.207
Compound **248**	0.076	6.566	0.147	−0.147	0.141
Compound **255**	0.072	6.901	0.156	−0.156	0.167
Compound **762**	0.063	7.880	0.153	−0.153	0.184
Dabrafenib	0.080	6.280 [38]	0.154	−0.154	0.149
**Compound**	**Electrodonating power** **(ω^-^)**		**Electroaccepting power** **(ω^+^)**		**Net Electrophilicity** **(Δω^±^)**
Compound **208**	0.299		0.132		0.432
Compound **248**	0.224		0.077		0.301
Compound **255**	0.254		0.098		0.352
Compound **762**	0.268		0.116		0.384
Dabrafenib	0.236		0.082		0.318 [38]

**Table 4 molecules-27-04098-t004:** Binding energies and Predicted binding affinities via Deep learning model.

Compound	Binding Energies (kJ/mol)	Predicted Binding Affinity (IC_50_) nM	pIC_50_ (Predicted via Deep Learning Model)
Compound **208**	−33.47	206.26	6.69
Compound **248**	−35.56	268.80	6.57
Compound **255**	−35.98	283	6.55
Compound **762**	−42.67	61.74	7.21
Dabrafenib	−33.89	1-9 (Experimental) [1]	---

**Table 5 molecules-27-04098-t005:** Average values obtained from MD simulations.

Protein-Ligand Complex	Average Protein RMSD (Å)	Average Protein RMSF (Å)	Average Protein-Ligand Complex RMSD (Å)	Average Radius of Gyration (Å)	Average SASA (Residue Wise) (Å^2^)
NEK7–Dabrafenib complex	1.97	0.87	3.89	19.76	282.72

**Table 6 molecules-27-04098-t006:** Important Hydrogen bonding observed during MD simulations.

Sr No.	Hydrogen Donor	Hydrogen Acceptor
1	ALA114-Main	LIGAND-Side
2	LIGAND-Side	LEU113-Side
3	LIGAND-Side	ASP115-Side
4	ARG121-Side	LIGAND-Side
5	LIGAND-Side	ASP179-Side
6	LIGAND-Side	GLU112-Main
7	GLY117-Main	LIGAND-Side
8	ASP115-Main	LIGAND-Side
9	LIGAND-Side	ALA114-Main
10	LIGAND-Side	ILE40-Main
11	LIGAND-Side	ARG42-Main
12	GLY41-Main	LIGAND-Side

**Table 7 molecules-27-04098-t007:** MM-GBSA binding energies of Dabrafenib docked at active site of NEK7.

	Binding Free Energy ΔG_bind_ (kcal/mol)	ΔE _coulomb_ (kcal/mol)	ΔE _covalent_ (kcal/mol)	ΔE _H-bond_ (kcal/mol)	ΔE _vdW_ (kcal/mol)	Lipophilic Energy (kcal/mol)	Sol_GB (kcal/mol)
Dabrafenib	−50.44	31.05	11.71	−0.18	−38.88	−33.98	−17.74

**Table 8 molecules-27-04098-t008:** MM-PBSA binding energies of Dabrafenib docked at active site of NEK7.

	Binding Free Energy ΔG_bind_ (kcal/mol)	ΔE _vdW_ (kcal/mol)	E_el_ (kcal/mol)	E_NPOLAR_ (kcal/mol)	E_PB_ (kcal/mol)	E_DISPER_ (kcal/mol)
Dabrafenib	−56.12	−38.27	−17.84	−26.71	32.37	47.53

**Table 9 molecules-27-04098-t009:** ADMET properties of top hits predicted via MPNN model.

Property Predicted	Compound 208	Compound 248	Compound 255	Compound 762
Solubility	−4.50 log mol/L	−4.05 log mol/L	−3.19 log mol/L	−3.10 log mol/L
Lipophilicity	1.82 (log-ratio)	1.89 (log-ratio)	1.40 (log-ratio)	1.88 (log-ratio)
(Absorption) Caco-2	−5.14 cm/s	−5.20 cm/s	−5.14 cm/s	−5.21 cm/s
(Absorption) HIA	91.18%	89.62%	89.69%	92.89%
(Absorption) Pgp	10.18%	13.10%	5.84%	11.42%
(Absorption) Bioavailability F20	76.34%	75.94%	75.46%	76.42%
(Distribution) BBB	76.85%	76.66%	93.85%	86.17%
(Distribution) PPBR	79.65%	77.11%	63.16%	80.66%
(Metabolism) CYP2C19	81.26%	72.00%	37.21%	24.15%
(Metabolism) CYP2D6	62.01%	51.90%	25.60%	13.64%
(Metabolism) CYP3A4	74.21%	56.97%	60.78%	22.83%
(Metabolism) CYP1A2	36.70%	10.29%	9.08%	21.05%
(Metabolism) CYP2C9	17.86%	6.99%	4.13%	8.60%
(Execretion) Half life	8.06 h	8.01 h	7.86 h	7.88 h
(Execretion) Clearance	8.23 mL/min/kg	8.26 mL/min/kg	8.10 mL/min/kg	8.54 mL/min/kg
Clinical Toxicity	14.92%	15.59%	24.33%	0.28%

## Data Availability

Not Applicable.

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
