# Peer review of "Deep Learning and Structure-Based Virtual Screening for Drug Discovery against NEK7: A Novel Target for the Treatment of Cancer"

_molecules, 2022, doi:10.3390/molecules27134098_

Round 1

Reviewer 1 Report

Comments to the authors:

In the paper, Aziz al. conducted a series of computational experiments including SBVS, DFT calculation, MD and deep learning to identify NEK7 inhibitor. Some major concern below:

1.Please use at least one experiment to verify the finding (i.e. NEK7 kinase assay), to see that is at least partially align with the computationally predicted values (using Dabrafenib as the control).

2.Please check the typo carefully throughout the manuscript.

i.e.  in abstract ‘small non-cell lung cancer’, do you mean non-small cell lung cancer?

NIMA (never in mitosis, gene A)-related kinase is NEK instead of ‘Never in Mitosis’

3. From Depmap, seems the knock out of NEK7 does not lead to cell killing, the author should give a better clarification how NEK7 inhibition can be used for cancer treatment in background,

Author Response

In the paper, Aziz al. conducted a series of computational experiments including SBVS, DFT calculation, MD, and deep learning to identify NEK7 inhibitors. Some primary concerns below:

  1. Please use at least one experiment to verify the finding (i.e., NEK7 kinase assay) to see that it partially aligns with the computationally predicted values (using Dabrafenib as the control).

Response: The reviewer gave a very fruitful suggestion and we agree that without experimental work justification of the results are partially validated. As this work belongs to the PhD project of Mr. Mubashir Aziz, we have already put purchase demand of our targeted Enzymes and cell-lines but purchasing is still in process and will take 2-3 month more. As experimental justification will take time and at this stage authors of this study are thankful in advance to the reviewer to consider molecular dynamics studies as justification for this point. We are working on more compounds and we will incorporate enzyme inhibition studies in our upcoming publication along with the justification of current compounds.

  1. Please check the typo carefully throughout the manuscript. i.e.,  In the abstract ‘small non-cell lung cancer,’ do you mean non-small cell lung cancer? NIMA (never in mitosis, gene A)-related kinase is NEK instead of ‘Never in Mitosis.’

 Response: Typo mistakes have been omitted throughout the manuscript as suggested.

  1. From Depmap seems the knockout of NEK7 does not lead to cell killing; the author should give a better clarification on how NEK7 inhibition can be used for cancer treatment in the background,

Response: A detailed clarification has been incorporated in the introduction part along with the references. NEK7 is a multifunctional protein. Atypical expression or any mutation may lead to the development of various cancers and inflammatory diseases. NEK7 not only promote the proliferation of resting cell but also mediate inflammatory pathways. Studies have proved the involvement of NEK7 in fatal inflammatory diseases like Atherosclerosis, Gout, liver cirrhosis, diabetic retinopathy, inflammatory bowel disease, rheumatoid arthritis, and tumor formation. So inhibition of NEK7 is crucial for eliminating NEK7-related malignancies.

Reviewer 2 Report

Manuscript Title:

A Comprehensive In-Silico Approach Based on Structure Based Virtual Screening and Deep Learning Approaches for Drug Repurposing and Drug Design: NEK7 a Novel Target for the

Treatment of Cancer

1. Why did the authors use both Structure Based Virtual Screening and Drug Repurposing? What is the specific reason for using both approaches?

2. Why did the authors use this specific library of benzene sulphonamide derivatives?

3. What is the specific reason for selecting a benzene sulphonamide scaffold?

4. Authors can improve the introduction section by adding the related references.

5. The authors can provide the URL and references for the databases and tools used in the present investigation.

6. I suggest authors carefully go through the manuscript to correct the mistakes related to grammar, punctuation, spelling and typos.

7. The authors had used 1ns for NVT and NPT equilibration steps? Is this time frame sufficient to attain the equilibration?

8. I suggest authors can perform an electrostatic surface potential map of all the ligand complexes to capture the nature of ligand binding in the drug target protein.

9. I suggest authors can perform following MD analysis of protein and protein-ligand complexes

- Radius of Gyration

- Intermolecular Hydrogen bonds

- Solvent Accessible Surface Area

- Essential dynamics based on PCA

Author Response

A Comprehensive In-Silico Approach Based on Structure-Based Virtual Screening and Deep Learning Approaches for Drug Repurposing and Drug Design: NEK7 a Novel Target for the Treatment of Cancer

  1. Why did the authors use both Structure-Based Virtual Screening and Drug Repurposing? What is the specific reason for using both approaches?

Response: Structure-based virtual screening was conducted to screen a compound library of benzene sulphonamide derivatives (1200 compounds) against NEK7. Whereas the drug repurposing is an advanced tool for revisiting the activities of already approved drugs for treatment against newer targets, the current study has employed drug repurposing for FDA-approved drug Dabrafenib against NEK7 as the treatment potential of Dabrafenib has not been previously explored against NEK7 associated cancer malignancies.

  1. Why did the authors use this specific library of benzene sulphonamide derivatives?

Response: The selection of the benzene sulphonamide library has two rationales:

Firstly, Dabrafenib, an FDA-approved protein kinase inhibitor, contains a benzene sulphonamide moiety as a basic pharmacophore in its structure, which results in a solid inhibiting potential against various proteins, such as an IC50 value in the nanomolar range (1-9 nM) against BRAF mutant myelomas. These myelomas have a significantly high level of NEK9 protein.

Secondly, the sulphonamide scaffold is present in numerous biologically active compounds, including anti-microbial, anti-tumor, anti-thyroid, antibiotics, and carbonic anhydrase inhibitors. In clinical practice, sulphonamide-possessing drugs treat lower urinary tract infections. The required references have also been incorporated in the revised manuscript.

  1. What is the specific reason for selecting a benzene sulphonamide scaffold?

Response: The sulphonamide scaffold is present in numerous biologically active compounds, including anti-microbial, anti-tumor, anti-thyroid, antibiotics, and carbonic anhydrase inhibitors. These inhibitors are used to treat infections and cancer malignancies. One potent FDA-approved protein kinase inhibitor (Dabrafenib) also possesses the same benzene sulphonamide scaffold. Owing to important biological activities, the current study has further explored the therapeutic action of the benzene sulphonamide scaffold against newer targets. 

  1. Authors can improve the introduction section by adding the related references.

Response: Corrected as per reviewer suggestion.

  1. The authors can provide the URL and references for the databases and tools used in the present investigation.

Response: The URL has been provided in the revised manuscript as per the reviewer's suggestion.

  1. I suggest authors carefully go through the manuscript to correct the mistakes related to grammar, punctuation, spelling, and typos.

Response: The authors have carefully corrected the mistakes related to grammar, punctuation, spelling, and typos as suggested.

  1. The authors had used 1ns for NVT and NPT equilibration steps? Is this time frame sufficient to attain the equilibration?

Response; Yes, after running the equilibration for 50,0000 steps (1ns), we have analyzed the trajectory and it was observed that system has attain the equilibration. (As shown in figure given below)

  1. I suggest authors can perform an electrostatic surface potential map of all the ligand complexes to capture the nature of ligand binding in the drug target protein.

Response: Electrostatic map potential for all the ligand complexes have been performed and incorporated in the manuscript as suggested.

  1. I suggest authors can perform following MD analysis of protein and protein-ligand complexes

- Radius of Gyration

- Intermolecular Hydrogen bonds

- Solvent Accessible Surface Area

- Essential dynamics based on PCA

Response: All the suggested calculations have been incorporated in the manuscript.

Reviewer 3 Report

In the present work, Aziz et al. perform in silico analysis looking for compounds able to inhibit NEK7 and using Dabrafenib as a reference since it is known to effectively inhibit NEK9.

This work has several major issues, and most of all, it provides only a few more data on the same topic already published in April 2022 by the same group of authors (10.1038/s41598-022-10253-5). 

TITLE

This is not a "drug repurposing" study. Drug repurposing is when new indications are found for drugs already used in clinical practice. Here, the authors analyze compounds that are not used in clinical practice, probably these molecules have not even undergone phase I clinical trial

ABSTRACT is excessively long and has little impact.

INTRODUCTION is poorly written, showing limited knowledge of cancer biology. For example (not exhaustive list):

- lines 55-56: "Breast cancer, liver.... are all kinds of cancer", this sentence is NOT acceptable in a scientific work

- lines 58-59: ".. all hallmarks of human cancer", these basic characteristics are typical of ALL tumors

- line 63: "Anti-mitotic medicines may be utilize...in the future", at present, there are dozens of anti-mitotic drugs currently used in clinical practice to treat tumors

Authors claim that they focus on NEK7 (line 75), but do not explain why. In lines 127-128, they state that this kinase is "associate with malignaces" which is far too vague information. It is sufficient to search PubMed for NEK7 + cancer to find lots of papers describing the role of this protein in different cancer settings. Since authors choose this target for their effort, they must rely on robust evidence to do so. And these references must be provided to readers.

From line 98 to line 120 authors describe their results, thus this part does not fit properly in the INTRODUCTION.

RESULTS

Many parts of the results section are just a description of the theory behind the tool used. This could be fine but need to be shortened. (lines 223-233, 248-260, 271-289)

Since from the beginning the RESULTS section, authors show results about 4 compounds without specifying from where they came. Probably from the 1200-compounds-from-PubChem (as stated in line 92) but it is never clearly specified, in particular, it is never specified how authors go from 1200 to 4. They state something in lines 307-309 but it is not sufficient and it must be explained at the beginning of the RESULTS.

In lines 466-467 and Table 5, authors comment on the "clinical toxicity" of compound 762. Since this is not assessable with in-silico tools, where did these data come from? 

CONCLUSION

Authors state that compound 762 "showed comparable binding affinity", related to Dabrafenib (line 479). Since these kinds of compounds are in the very early phases of in-vitro testing, no phase I clinical trial has been performed and no information about pharmacokinetics and pharmacodynamics is available. Conversely, Dabrafenib is widely used in clinical practice. So, which is the benefit of recommending using this compound to target NEK7 when its efficacy is comparable with Dabrafenib? Moreover, the same analysis and conclusion have yet been made by the authors in their paper published in April 2022.

Author Response

This work has several significant issues, and most of all, it provides only a few more data on the same topic already published in April 2022 by the same group of authors (10.1038/s41598-022-10253-5). 

Response; we are thankful to reviewer comments for spotting revisions and helping in improving our study. Moreover, we are grateful to the reviewer in advance for accepting the modifications.

Our previous study (10.1038/s41598-022-10253-5) focused on evaluating the inhibiting potential of Phenylcarbamoylpiperidine-1,2,4-triazole amide derivatives against NEK7. Our research group synthesized these derivatives, but the current research is a different where in-silico approach based on SBVS and Deep learning models for investigating the library of 1200 compounds against NEK7 was designed. We have incorporated some data from our previous study, and due reference has been added.

TITLE

This is not a "drug repurposing" study. Drug repurposing is when new indications are found for drugs already used in clinical practice. Here, the authors analyze compounds that are not used in clinical practice. Probably these molecules have not even undergone phase I clinical trial

Response: Corrections have been made as suggested. In the current study, a Drug repurposing approach was employed for FDA drug Dabrafenib whereas SBVS was conducted for the compound library to search more better inhibitor that can form the basis of synthesis of more potent inhibitors of NEK7 and luckily we got three compounds for further investigations.

ABSTRACT is excessively extended and has little impact.

Response: The abstract has been shorted and has a substantial impact.

INTRODUCTION is poorly written, showing limited knowledge of cancer biology. For example (not exhaustive list):

Response; Introduction has been significantly improved to suit the research scope as suggested.

- lines 55-56: "Breast cancer, liver.... are all kinds of cancer" this sentence is NOT acceptable in a scientific work

Response; Corrections have been made as suggested

- lines 58-59: ".. all hallmarks of human cancer" these fundamental characteristics are typical of ALL tumors

Response; Corrections have been made as suggested

- line 63: "Anti-mitotic medicines may be utilized...in the future" at present, there are dozens of anti-mitotic drugs currently used in clinical practice to treat tumors

Response; Corrections have been made as suggested

Authors claim that they focus on NEK7 (line 75) but do not explain why. In lines 127-128, they state that this kinase is "associated with malignancies," which is far too incomplete information. It is sufficient to search PubMed for NEK7 + cancer to find lots of papers describing the role of this protein in different cancer settings. Since authors choose this target for their effort, they must rely on robust evidence. And these references must be provided to readers.

Response; As suggested, we have incorporated changes in the revised manuscript. Briefly, we have searched PubMed for NEK7 role in cancer, and other inflammatory conditions and sufficient data has been incorporated in the introduction part. More than five papers have been used to describe the role of NEK7, and due reference has been added.

From line 98 to line 120, the authors describe their results. Thus this part does not fit appropriately in the INTRODUCTION.

Response; Corrections have been made as suggested

RESULTS

Many parts of the results section just describe the theory behind the tool used. This could be fine but need to be shortened. (lines 223-233, 248-260, 271-289)

Response; Corrections have been made as suggested.

Since from the beginning the RESULTS section, authors show results about 4 compounds without specifying from where they came. Probably from the 1200-compounds-from-PubChem (as stated in line 92) but it is never clearly specified, in particular, it is never specified how authors go from 1200 to 4. They state something in lines 307-309 but it is not sufficient and it must be explained at the beginning of the RESULTS.

Response; Corrections have been made as suggested. Explanation has been provided at the beginning of the results.

In lines 466-467 and Table 5, the authors comment on the "clinical toxicity" of compound 762. Since this is not assessable with in-silico tools, where did these data come from? 

 Response; we have predicted clinical toxicity value via MolDesigner, a fantastic tool for designing efficacious drugs with a deep learning model. This tool uses the MPNN model for the prediction of ADMET properties. The research paper and Codes details are available at https://arxiv.org/abs/2010.03951. Moreover, references to deep learning models are already incorporated in the manuscript.

CONCLUSION

The authors state that compound 762 "showed comparable binding affinity", related to Dabrafenib (line 479). Since these kinds of compounds are in the very early phases of in-vitro testing, no phase I clinical trial has been performed and no information about pharmacokinetics and pharmacodynamics is available. Conversely, Dabrafenib is widely used in clinical practice. So, which is the benefit of recommending using this compound to target NEK7 when its efficacy is comparable with Dabrafenib? Moreover, the same analysis and conclusion have yet been made by the authors in their paper published in April 2022.

Response; Yes, the reviewer is correct; these compounds are in the early stages of drug development; our in-silico-based approach contributed toward investigating the inhibiting potential via molecular docking and determining the drug-like properties of these compounds. We maintained Dabrafenib as a standard drug to which our in-silico findings could be compared. Our study revealed that compound 762 has an excellent binding affinity with NEK7. Moreover, it also possesses drug-like properties (ADMET profile). These findings will assist researchers in developing newer leads without consuming much time and money.

Round 2

Reviewer 1 Report

The authors address most of the concerns.

Author Response

Comments and Suggestions for Authors

Comment: The authors address most of the concerns.

Response: All authors extend their sincere gratitude to the reviewer for accepting the revisions.

Reviewer 2 Report

  1. The authors have to elaborate on the results and discussion section of the electrostatic surface potential map calculation of all the complexes.
  2. The authors can prepare the SASA figure based on residue wise. The authors can also compute buried surface area for the best complexes.
  3. The authors can introduce one more table which explains the average values obtained from MD simulation of free protein and ligand bounded complexes. This table includes the average value of RMSD, RMSF, Rg, Trace covariance matrix, etc. 
  4. In the MM-GBSA table, the binding free energy column is missing.
  5. Why did the authors not perform MM-PBSA-based binding free energy analysis? Why did the authors choose MM/GBSA based? Is there any specific reason?

Author Response

Comments and Suggestions for Authors

  1. The authors have to elaborate on the results and discussion section of the electrostatic surface potential map calculation of all the complexes.

Response: Authors expresses sincere gratitude to reviewer for suggesting changes to improve the quality of manuscript.

The result section of electrostatic surface map potential is elaborated as suggested.

  1. The authors can prepare the SASA figure based on residue wise. The authors can also compute buried surface area for the best complexes.

Response: The residue based SASA figure has been incorporated in the manuscript. In addition, comprehensive Buried surface area for best complexes with extensive elaboration has been incorporated as per suggestions.

  1. The authors can introduce one more table which explains the average values obtained from MD simulation of free protein and ligand bounded complexes. This table includes the average value of RMSD, RMSF, Rg, Trace covariance matrix, etc. 

Response: Table has been incorporated as suggested.

  1. In the MM-GBSA table, the binding free energy column is missing.

Response: It was typo mistake which has been corrected.

  1. Why did the authors not perform MM-PBSA-based binding free energy analysis? Why did the authors choose MM/GBSA based? Is there any specific reason?

Response: We have performed both MMPBSA and MMGBSA based binding free energy analysis, but we have included MMGBSA analysis because MMGBSA results are relatively stable with high precision. Moreover, MMGBSA results are reproducible as they are insensitive to different variations in solvent model, ligand and protein structural change, and uncertainty in protonation state. MMGBSA can be used for a long length of MD simulation. Whereas the performance of MMPBSA depends upon the solvation model and test system. Moreover, MMPBSA ignores the structure changes of the ligand and the receptor on the ligand binding. Its calculation is based on single minimised structures rather than a large number of MD snapshots, which ignore the dynamic effect. PBSA has poor precision and a high standard deviation.

However, as per suggestion of reviewer, we have incorporated the results of MMPBSA in the manuscript.

Reference to above MMGBSA/MMPBSA discussion is provided below;

Genheden, Samuel, and Ulf Ryde. “The MM/PBSA and MM/GBSA methods to estimate ligand-binding affinities.” Expert opinion on drug discovery vol. 10,5 (2015): 449-61. doi:10.1517/17460441.2015.1032936

Reviewer 3 Report

This work has several significant issues, and most of all, it provides only a few more data on the same topic already published in April 2022 by the same group of authors (10.1038/s41598-022-10253-5). 

Response; we are thankful to reviewer comments for spotting revisions and helping in improving our study. Moreover, we are grateful to the reviewer in advance for accepting the modifications.

Our previous study (10.1038/s41598-022-10253-5) focused on evaluating the inhibiting potential of Phenylcarbamoylpiperidine-1,2,4-triazole amide derivatives against NEK7. Our research group synthesized these derivatives, but the current research is a different where in-silico approach based on SBVS and Deep learning models for investigating the library of 1200 compounds against NEK7 was designed. We have incorporated some data from our previous study, and due reference has been added.

Thank to the authors for this explanation and for including the reference. Nevertheless, their previous work has been added only as a reference for the “Guassian09 program” (page 4, line 351). Also readers will benefit from the clarification authors kindly provided here. I encourage them to add this information and reference also in the final manuscript, maybe in the Conclusion section.

---------------------------------------------------------------------------------------------------------------------The authors state that compound 762 "showed comparable binding affinity", related to Dabrafenib (line 479). Since these kinds of compounds are in the very early phases of in-vitro testing, no phase I clinical trial has been performed and no information about pharmacokinetics and pharmacodynamics is available. Conversely, Dabrafenib is widely used in clinical practice. So, which is the benefit of recommending using this compound to target NEK7 when its efficacy is comparable with Dabrafenib? Moreover, the same analysis and conclusion have yet been made by the authors in their paper published in April 2022.

Response; Yes, the reviewer is correct; these compounds are in the early stages of drug development; our in-silico-based approach contributed toward investigating the inhibiting potential via molecular docking and determining the drug-like properties of these compounds. We maintained Dabrafenib as a standard drug to which our in-silico findings could be compared. Our study revealed that compound 762 has an excellent binding affinity with NEK7. Moreover, it also possesses drug-like properties (ADMET profile). These findings will assist researchers in developing newer leads without consuming much time and money.

 I appreciate this specification, but as above, also readers would benefit from this clarification. Authors should include this discussion in the Conclusion section.

----------------------------------------------------------------------------------------------------------------

All other issue has been sufficiently addressed. Thank to the authors for their efforts. 

Author Response

Comments and Suggestions for Authors

  1. Thank to the authors for this explanation and for including the reference. Nevertheless, their previous work has been added only as a reference for the “Guassian09 program” (page 4, line 351). Also readers will benefit from the clarification authors kindly provided here. I encourage them to add this information and reference also in the final manuscript, maybe in the Conclusion section.

Response. Thank you for appreciation and accepting our revisions. We have incorporated reference of our previous study in result and discussion part as per suggestions. Moreover suggested information is also incorporated in conclusion part.

  1. The authors state that compound 762 "showed comparable binding affinity", related to Dabrafenib (line 479). Since these kinds of compounds are in the very early phases of in-vitro testing, no phase I clinical trial has been performed and no information about pharmacokinetics and pharmacodynamics is available. Conversely, Dabrafenib is widely used in clinical practice. So, which is the benefit of recommending using this compound to target NEK7 when its efficacy is comparable with Dabrafenib? Moreover, the same analysis and conclusion have yet been made by the authors in their paper published in April 2022.

 I appreciate this specification, but as above, also readers would benefit from this clarification. Authors should include this discussion in the Conclusion section.

Response:  we have incorporated suggested discussion and clarification in conclusion part as per suggestion of the reviewer.

  1. All other issue has been sufficiently addressed. Thank to the authors for their efforts. 

Response: we are highly in debt and grateful to reviewer for suggesting changes to improve quality of our work.